# An API for Wearable Environments Development and Its Application to mHealth Field [note 1]

**DOI:** 10.3390/s20215970

**Published:** 2020-10-22

**Authors:** Fabio Sartori

**Affiliations:** Department of Informatics, Systems and Communication, University of Milano-Bicocca, 20126 Milano, Italy

**Keywords:** Android Wear, Bluetooth Low Energy, application configuration, wearable/mobile sensor information, smart environments, Internet of Things, mHealth

## Abstract

Wearable technologies are transforming research in traditional paradigms of software and knowledge engineering. Among them, expert systems have the opportunity to deal with knowledge bases dynamically varying according to real-time data collected by position sensors, movement sensors, etc. However, it is necessary to design and implement opportune architectural solutions to avoid expert systems are responsible for data acquisition and representation. These solutions should be able to collect and store data according to expert systems desiderata, building a homogeneous framework where data reliability and interoperability among data acquisition, data representation and data use levels are guaranteed. To this aim, the wearable environment notion has been introduced to treat all those information sources as components of a larger platform; a middleware has been designed and implemented, namely WEAR-IT, which allows considering each sensor as a source of information that can be dynamically tied to an expert system application running on a smartphone. As an application example, the mHealth domain is considered.

## 1. Introduction

Wearable technology has the ability to provide applications with functionalities typically not present in mobile and laptop devices. Examples of these functionalities are monitoring of physiological functions, evaluation of environmental variables, etc. A number of smart applications can obtain benefits from employing wearable devices; in particular, being important sources of data for reasoning, they could be used in the design and implementation of a new generation of expert systems.

As reported in [1], developing expert systems and, more generally, intelligent systems, is a complex task, where both deep understanding of the problem domain and the definition of heterogeneous knowledge representation techniques and tools are needed. These techniques are usually based on different paradigms and platforms.

In this framework, Internet of Things (IoT) has offered new and important possibilities for the design and implementation of expert systems; currently, every device can be connected to the Internet and provide valuable information for decision-making [2]. In the well-known tripartite vision of IoT [3], expert systems are located in the semantic-oriented area, given that they have been always conceived as applications for reasoning about data; expert systems should be able to distribute knowledge bases, in order to gather data from sparse resources maximizing the benefits for their reasoning strategies.

In a previous paper [4], the wearable expert systems (WES) notion is introduced as an innovative paradigm to develop knowledge based systems able to modify deliberations dynamically, according to the temporal evolution of data acquired from wearable devices. In that paper, the design and implementation of WES from the knowledge engineering perspective is faced, focusing on the conceptual model necessary to develop a complete rule-based system from a bottom-up analysis of the relations among the data collected in the environment. In other words, data were assumed to be perfectly reliable and always available, since the final goal was to test the capability of the WES framework to develop a correct reasoning process depending on a variable set of observations.

In this paper, the goal is to show how the wearable expert systems can be connected to real-world wearable devices in order to build effective mobile applications. To this aim, it is important to notice that the data feeding the system are characterized by several parameters defining their suitability to applications. For this reason, the WES notion has been extended in order to give a complete view of modern mobile applications, defining the wider concept of *Wearable Environment*.

According to [5], the service-oriented vision considers IoT as “the inter-networking paradigm enabled by technology stack which provides a seamless connectivity between physical and virtual objects to facilitate the development of intelligent services and applications with self-configuring capabilities.” A recent paper [6] on *sustainability of wearable devices* points out that the development of middleware technology is crucial for the design and implementation of services that support users through wearable devices. The main reason is the need for heterogeneous systems to co-exist into a unique conceptual framework.

In this context, Wearable Environment Acquisition and Representation InfrasTructure (WEAR-IT) is a collection of APIs to connect data gathered by wearables and WESs. The goal of WEAR-IT is to build an overall view of a knowledge domain, where data are acquired from sets of wearables and applications can choose the best data sources on the basis of their needs. This API layer allows the dynamic connection of data sources to distributed applications. An App, accessing the WEAR-IT set of APIs, can be hosted on a large number of smartphones and interrogated about the device capabilities and the willingness of the owner to contribute certain data by periodically publishing sensor measurements.

## 2. Materials and Methods

### 2.1. Related Work

Much work has been carried out in recent years on the definition, specification and deployment of IoT infrastructures (see, for instance, the following surveys [7,8]). However, the general problem of realizing open, application-neutral platforms did not attract much attention, and it is far from having found a satisfactory solution.

For instance, some proposals aimed at achieving interoperability at the protocol level privilege, based on interworking gateways, overlooking the convenience of a common programming environment [9]. The many works stemming from the Web of Things model (see, for instance, [10,11,12]) provide solutions to the information semantics problem, but they assume implicitly rather complex, resource-rich, endpoint implementations. Moreover, they are not well suited to situations where the applications may be either centralized or (fully or partially) distributed to the endpoints; in our view, providing a programming model supporting any application distribution strategy would be beneficial. Finally, the issues of searching for the needed sensors and enrolling and configuring them on-the-fly are almost absent from the literature.

As reported in [13], wearable sensors have become very popular in many applications, being useful in providing reliable and accurate information on people’s behavior. Most of these applications are related to the medical [14,15,16] and sport and training [17,18,19] domains.

Ubiquitous healthcare systems [20] exploit many hardware and software components, e.g., wireless sensor networks [21] and wireless body area networks [22]. Moreover, they typically use mobile devices and wireless cloud services, with the aim to reach pervasive availability.

Indeed, the mHealth domain is one of the most interesting and promising fields of application of wearable technologies. Despite this, they are still underutilized in the healthcare domain [23]. In [24], four main causes are pointed out; in particular, the *lack of interoperability and standardization* in detecting and storing acquired data is considered very significant. In this field, great efforts have been made to assess the accuracy of wearable sensors in classifying *activities of daily living* (ADL). The utility of accelerometers to evaluate the performance of ADL by elderly people monitored at home has been demonstrated [25]. A prototype system using in-shoe pressure and acceleration sensor has been proposed [26] to classify activities such as *sitting*, *standing* and *walking*. Patients affected by degenerative disorders, such as Parkinson’s disease, are an important target: in [27] an accelerometer-based device designed for step counting in patients with Parkinson’s disease is presented. Wearable sensors were used in [28] to control the rehabilitation of patients at home after abdominal surgery.

The important facilitating role of wearable technologies for the promotion of a healthy lifestyle is stressed in [29]. Obese individuals support and the implementation of clinical interventions based on promoting an active and healthy lifestyle through wearable technologies have been proposed [30,31]. Monitoring physiological data can improve the diagnosis and treatment of chronic conditions; e.g., cardiovascular diseases can benefit from continuous monitoring of physiological parameters [32].

This paper presents an attempt to overcome the limitations of the architecture of the applications above, which is based on wireless sensor networks communicating with fixed stations where data are collected and managed. The approach is very similar to the principles of *wireless sensor networks with mobile elements (WSNME)* proposed in [33]. In WSNME, special support nodes are introduced in addition to sensor nodes and information sinks: these nodes act as intermediate data collectors or mobile gateways.

As reported in [34], mobility in WSNs is useful for several reasons, e.g., increased reliability and reduced cost of data transmission, connectivity benefits and energy efficiency. The main difference between WSNME and traditional applications such as body sensor networks is the introduction of this intermediate layer, where data from sensors are collected, making them available to interested users. In our model, interested users are wearable expert systems and the intermediate level is implemented by smartphones communicating with an IoT platform; this approach leads to the following definition of a wearable environment.

### 2.2. Definition of Wearable Environment

A *wearable environment* is a triple
WE={A,MD,S}
where
$A=\{a_{1},a_{2},\cdots ,a_{n}\}$ is a set of *applications*, possibly interconnected;$MD=\left\{wd_{1},wd_{2},\cdots ,wd_{m}\right\}\cup \left\{sp_{1},sp_{2},\cdots ,sp_{k}\right\}$ is a set of *wearable devices* and *smartphones*, possibly interconnected; and$S=\{s_{1},s_{2},\cdots ,s_{t}\}$ is a set of *sensors*.

Figure 1 shows how a wearable environment can be characterized in terms of the wearable devices involved. Nowadays, wearable devices can cover the entire surface of a human body: in particular, *smart-wristbands and smart-watches* are very important from the mHealth perspective, since they can record physiological parameters such as heart rate, calories burned during the day, etc; they are usually equipped with movement tracking functions that keep trace of distance walked and the amount of time active.

In our definition of a wearable environment, the interconnection among all these devices is mediated by a *smartphone*, which plays two roles, being both a data-generating device and a data collection hub for other wearables. For a large-scale application, the wearable environment would include several smartphones, each acting as the hub for a group of wearables: in the following, we describe the simplest case of a single smartphone, although the extension to a larger environment is possible.

### 2.3. WES Level: Applications

Applications at reasoning level of a wearable environment are WES, typically characterized as rule-based systems. They implement a specific decision making process exploiting the variability of the related knowledge bases over the time [35,36]. This is due to the exploitation of information sources able to gather data from the environment around; the observed system and its reference environment evolve through a series of macroscopic states, each characterized by a specific set of relevant rules.

The transition from one state to the next causes that some events can become less important than others; therefore, the inferences activated by such events should decrease their salience accordingly. To take care of such dynamic evolution of WES knowledge bases, *bayesian networks* [37] have been integrated into the WES specification, to identify, for each time frame, the set of rules fitting better with the current macroscopic state, on the basis of a specific strategy [38].

### 2.4. API Level: A Bridge between Data and Applications

Applications at reasoning level could adopt different paradigms to implement their decision making processes. Thus, they should be able to interact with data in the most profitable way accordingly. This issue implies rethinking the relationship between applications and wearable devices providing data, enabling the former to select, browse and query the preferred device according to their goals.

Figure 2 depicts this change of perspective. The traditional two-tier architecture of a wearable expert system, based on *acquisition* of data from sensors on a wearable device and *reasoning* exploiting them, is extended to a three-tier architecture by adding an explicit *data representation* level. The main benefit is the possibility to make the reasoning process independent of the specific representation of necessary data: while in the first case (on the left in Figure 2) the WES must implement an opportune strategy for representing information gathered by sensors, in the second one it must only notify the representation level about the characteristics of needed data and their format, being notified when data are available accordingly.

To reach this objective, a specific framework has been developed, namely *Wearable Environment Acquisition and Representation InfrasTructure* (WEAR-IT). A complete API makes the reasoning level able to communicate with the data one, which is described in the following section.

## 3. The WEAR-IT API

### 3.1. Interpreting WEs as FSMs

Figure 3 presents a wearable environment as a Mealy [39] finite state machine (FSM):WE={I,U,S,f,g}
where the output state depends on the current state of the machine and the input received by it:I={Scan,Connect,Sensors,Classify,Select,Play,Stop,Store} is the *input alphabet*, made of primitives that can activate transitions of states.U={⌀,WearableDevice,Sensor,True,False,Path} is the *output alphabet*.S={Reasoning,Representation,Acquisition} is the finite set of possible states; in particular, the initial state $S_{0}=Reasoning$;.f:I×S⟶S: is the *transition function*S(t+1)=f(I(t),S(t)), mapping pairs of a state and an input symbol to the corresponding next state.g:I×S⟶U: is the *output function*U(t)=g(I(t),S(t)), mapping pairs of a state and an input symbol to the corresponding output symbol.

This schema is useful to understand the role of the *representation* level in the wearable environment architecture, as well as the characterization of a wearable environment as a whole composed of heterogeneous entities.

When an application, i.e., a WES, is activated on the smartphone, the wearable environment enters the *reasoning* level. This is the initial state of the FSM: in fact, the information flow within a wearable environment is always started according to the needs of one or more applications. It is important to notice that the *reasoning* level is the final state too: the application could decide to execute its own decision making process and terminate without interacting with other levels, depending on its strategy. Otherwise, it will move to the *representation* level to obtain necessary data to complete the body. This state is activated by a *scan* input, which specifies the application needs to know which wearable devices and sensors are available nearby to gather data. From now on, the wearable environment can maintain the representation level, move to *acquisition* level or come back to *reasoning* level. The *sensors* and *classify* inputs cause the level maintenance, being actions that provide lists of sensors around and their clustering according to the nature of detected data; the *connect*, *select*, *play* and *stop* inputs cause the transition to the *acquisition* state, being actions that allow establishing a connection with a specific wearable device, selecting one sensor on it and starting/stopping detection of data through it. Finally, coming back to the *reasoning* state is performed when the *store* input is given, since all the data necessary to the reasoning level to accomplish its actions are available.

Note that the transition from *acquisition* level to the *representation* one is automatic, at the end of the required primitive. Table 1 resumes the state transition table of a wearable environment.

Table 2 presents the output function *g* for each state. The Scan transition from *reasoning* to *representation* returns a list of *wearable devices*. During the persistence in the *representation* state many transitions occur. Towards the *acquisition* state, the Connect input returns the identification of a wearable devices among the previously scanned; the Sensors one allows obtaining a (possibly empty) *list of sensors* the previously connected wearable devices is equipped with; the Classify transition arranges such sensors in a tree-like structure according to their nature; the Select input allows deciding which sensor to query; and the Play/Stop actions start/terminate the acquisition of data from the selected sensor returning True/False values on the basis of sensor availability. Finally, the Store transition towards the *reasoning* state provides the *path* to the data storage. As introduced above, the transitions from *acquisition* state to the *representation* one are automatic at the end of the necessary operations. Since the goal of acquisition state is to finalize the interaction with selected wearable device or sensor, the result of such finalization is returned as a boolean value.

### 3.2. Operational Semantics

The last step of the wearable environment API formalization is the definition of an operational semantics to detail the interactions among the states. This is necessary to clearly define which kinds of services can be invoked by a state. Each primitive is represented as a *method* (see Table 3) characterized by:**Name**: the name of the method, to identify the service in a non ambiguous way.**Parameters**: the (possibly empty) list of values necessary to perform the required service.**Description**: a brief explanation of the method body.**Return value**: the value obtained by the service invoking entity at the end of the method. This value must be coherent with Table 2.

#### 3.2.1. *Scan*()

This function searches all the wearable devices wd within a wearable environment that can be associated to a smartphone sp (see Table 4).

#### 3.2.2. *Connect(sp, wd)*

This function (see Table 5) returns *true* if the pairing operation between the smartphone sp and one of the wearable devices wd previously listed by Scan() succeeds, *false* otherwise. Pairing is made thanks to $Bluetooth^{\mathrm{TM}}$ technology.

#### 3.2.3. *Sensors(sp)* and *Sensors(wd)*

These methods (see Table 6) enable the invoking application to obtain a complete list of sensors available on the smartphone sp it is running on, as well as a previously connected wearable device wd.

#### 3.2.4. *Classify(sp)* and *Classify(wd)*

These primitives (see Table 7) group all sensors on the smartphone sp where the invoking application runs and/or on the wearable device wd, already paired with sp, according to a hierarchical structure. This structure, depicted in Figure 4, complies with $Android^{TM}$ categorization of sensors in *position sensors*, *movement sensors* and *environment sensors*.

#### 3.2.5. *Sensor(sp)* and *Sensor(wd)*

The collection of sensors List<Sensor> returned by Sensors primitive depends on the device characteristics. It is extremely rare that a device is equipped with the totality of sensors available on the market. For example, *accelerometer*, *magnetometer* and *gyroscope* are common, while others can be derived from them (e.g., the rotation vector).

Since sensors could deprecate their performance due to accidents and/or aging, it is very important for an application to know if a specific sensor is working, even though it has been listed in the Sensors(sp) and/or Sensors(wd) result. Sensor(sp,s) and Sensor(wd,s), shown in Table 8, return *true* if the sensor *s* is really available for data gathering on a smartphone sp and/or a wearable device wd.

#### 3.2.6. *Select(sp, s)* and *Select(wd, s)*

Once the wished sensor *s* has been identified and verified by the application, through the Sensors(sp,s)/Sensors(wd,s) and Sensor(sp,s)/Sensor(wd,s) primitives, respectively, the Select() function in Table 9 allows activating *s* on sp or wd:

The method returns information about the sensor to the invoking application. In particular, the *sensor identifier* (e.g., the UUID) is exploited by the reasoning level of the wearable environment for the next operations on it.

#### 3.2.7. *Play(sp)* and *play(wd)*

The Play() function is the most important part of the WEAR-IT API: given a sensor *s*, which has been previously identified, verified and selected on a smartphone sp and/or wearable device wd, it allows querying the sensor at different levels of granularity, in order to obtain detection of data from it, according to the reasoning level specification.

Table 10 shows the most extended interface of this function: the method can be properly overloaded according to the needs of the application. The parameters required are:**Frequency f**: This is the sampling frequency to collect data from the sensor *s*. The default setting is 5000 milliseconds.**Time period [$t_{0}$, $t_{n}$]**: This is the time interval during which the application wishes to collect data from sensor *s* at the given frequency f. Typically, $t_{0}$ is the instant when the primitive is invoked and $t_{n}>t_{0}$; if $t_{0}=t_{n}$ only one value will be instantaneously returned, while $t_{n}<t_{0}$ will generate an error.

Possible variants are Play(sp,s,n)/Play(wd,s,n), which allows obtaining *n* consecutive readings of sensor *s* value on a smartphone sp and/or a wearable device wd, sampled at the default frequency and Play(sp,s)/Play(wd,s), which simply reads a value from sensor *s* when invoked.

#### 3.2.8. *Stop(sp, s)* and *Stop(wd, s)*

This primitive interrupts the data collection from a sensor *s*, being it on a smartphone sp or a wearable device wd. Table 11 resumes it. This function can only be used if Play() has been previously invoked. The following cases can verify:If Play(*, s) has been invoked, with * = sp or * = wd, then *Stop(*, s)* is not mandatory.If Play(*, s, $t_{0}$, $t_{n}$, f) has been activated, with * = sp or * = wd and $t_{0}<t_{n}$, then Stop(s) can be used to terminate the execution of Play(*, s, $t_{0}$, $t_{n}$, f) before the normal exit at the end of the $\left[t_{0},t_{n}\right]$ time period.

#### 3.2.9. *Store*()

Given a list of values, possibly composed of only one element, acquired by a previous Play() call, the Store() primitive allows archiving the data under different formats, such as JSON, XML, CSV, etc. As per the others, the *Store*() function aims to provide the applications at reasoning level of a wearable environment with a set of configurations to meet their needs. The default settings for parameters are the following ones:**File name**: a string composed by the *name of the sensor* and a *timestamp*, joined by “_”.**Archiving directory**: the default directory of the application within the device where it runs.**Format**: the format of the file may depend on the type of data gathered by the sensor; the JSON (or XML) format is used to standardize it.**Access mode**: data are generally *appended* to the file, in order to preserve possible existing information.

Each variant of Store() returns the *path* of the file according to the operation result. In the case of errors, an error code is generated. In the case of success, the returned value could be an absolute path within the device (if default settings are active) or an *URI/URL* in the case of distributed resources. Table 12 summarizes the most extended interface of the Store() method.

### 3.3. An Example

WEAR-IT has been developed under $Android^{TM}$ OS, since many wearable devices fully compatible with the $AndroidWear^{TM}$ interface are available at low cost on the market. Android OS provides a collection of primitives through which the sensors of a smartphone, or an Android Wear device, can be queried.

While developing WEAR-IT, other methods of interconnections among wearable devices have been considered, focusing on $BluetoothLowEnergy^{TM}$ technology. Other possibilities are the subject of future works (see Section 6).

Algorithm 1 shows a sketch of the usage of the WEAR-IT APIs. As shown in Figure 1, the smartphone sp is the only input necessary. Through the *scan* operation, the smartphone is able to determine the set of other available wearable devices, showing them to the user, which can be either a human being (if the graphical user interface is employed) or a software application. The user is then enabled to choose one specific wearable device wd, by means of the *connect* primitive; wd can then return the list of sensors it is equipped with and the user can select one of them exploiting the *select* operation. Finally, the *play* and *stop* functions are invoked between two distinct time instants $t_{1}$ and $t_{n}$, in order to detect raw data from the sensor for the desired period of time. As reported above, if $t_{1}=t_{n}$, the $data_{s}$ variable contains a single value.
**Algorithm 1** WEAR-IT API.**Require:**sp**Ensure:**$s,wd,data_{s}$     $\left\{wd_{i}\right\}=scan\left(sp\right),i\in \left[1\ldots k\right]$     connect to a device wd suitable for the application     connect(wd,sp)     identify the sensors in the wearable environment     $\left\{s_{j}\right\}=classify\left(sensors\left(sp\right)\cup sensors\left(wd\right)\right),j\in \left[1\ldots m\right]$     select the needed sensor of type T     $s=select(\left\{s_{j}\right\},T)$     play(s)     measurement takes place     $data_{s}=stop\left(s\right)$     **return**
$store(data_{s},wd)$

An interesting extension of the algorithm above would be the remotization of WEAR-IT through the integration of IoT platforms. This would allow implementing dynamically the communication component of a smart city application. To provide a complete support to this kind of applications, it is possible to extend the API with the primitives needed to identify and select the endpoints suitable to the specific application needs. The primary extension is a search(area,N) primitive which, in its simplest form, returns a list of up to N smartphones supporting WEAR-IT, presently located within the area boundaries. As an example, consider an application which requires a snapshot of the climate conditions (temperature, pressure and humidity) in a specific area. The implementation could follow the simple schema sketched in Algorithm 2.
**Algorithm 2** Distributed data collection with WEAR-IT.$\left\{sp_{i}\right\}=search\left(area,N\right)$**for all***i***do**      $\left\{s_{j}\right\}=sensors\left(sp_{i}\right)$      $st=select(\left\{s_{j}\right\},TEMPERATURE)$      **if**
st<>NULL
**then**             t=play(st)      **end if**      $sr=select(\left\{s_{j}\right\},PRESSURE)$      **if**
sr<>NULL
**then**             p=play(sr)      **end if**      $sh=select(\left\{s_{j}\right\},HUMIDITY)$      **if**
sh<>NULL
**then**             h=play(sh)      **end if**      **if**
t<>NULL
**then**             $store(sp_{i}.st,t)$      **end if**      **if**
p<>NULL
**then**             $store(sp_{i}.sr,p)$      **end if**      **if**
h<>NULL
**then**             $store(sp_{i}.sh,h)$      **end if****end for**

Figure 5 shows how Algorithm 1 works in practice, by means of an $Android^{TM}$ app developed to test the proposed API.

Here, it is supposed that
$${WEAR-IT}_{wd}=\left\{sp\cup \left\{WearOS_{i}\right\}\cup \left\{BLE_{j}\right\}\right\}$$
with j∈[1…n] and j∈[1…m], being possible that $\{WearOS_{i}\}=\emptyset$ and $\{BLE_{j}\}=\emptyset$. At launch, the application allows scanning the wearable devices around the host smartphone and choose the device to connect with; this device belongs to one of the ${WEAR-IT}_{wd}$ subsets (Steps 1 and 2 in Figure 5). Then, the *classify* primitive is invoked (Step 3 in Figure 5). The wd sensors are clustered into four categories:*Sensor available* provides the list of all sensors mounted on the wearable device, alphabetically ordered. This function is useful to have a quick view about all the possible data an application at the reasoning level can exploit from the current device. The list of sensors can be exported to be used by applications.*Motion, environment and position sensors* group the sensors involved in the detection of the values necessary for the correct execution of an application, on the basis of its goals. The sensors (e.g., accelerometer) belonging to *motion* category can be exploited by applications interested in the analysis of the user movement, e.g., recommender systems for training. The sensors (e.g., light) belonging to the *environment* category can be interesting for applications suggesting actions to take in response to changes in the wearable environment context, e.g., personalized entertainment. The sensors (e.g., orientation) belonging to *position* category can be queried by applications interested in the analysis of the user geographic position, e.g., systems for suggesting places to eat.

The *select* primitive usage is shown as Step 4 in Figure 5; accessing a category returned by *classify*, it is possible to watch the list of clustered sensors. Finally, raw data can be acquired and permanently stored by means of *play*, *stop* and *store* functions (Steps 5 and 6 in Figure 5). The final configuration of the wearable environment obtained at the end of the algorithm in also shown.

## 4. Case Study

Figure 6 shows a concrete implementation of the wearable environment conceptual model (see Figure 3 into a computational model, where an mHealth app, namely *MoveUp*, is considered at reasoning level. The diagram presents all the interactions among the different components of the wearable environment as well as interactions with external entities, such as the *storage*, allowed by the primitives previously described.

Looking at this diagram, the wearable environment is a platform where *software components*, i.e., wearable expert systems), *hardware components*, such as smartphone, wearable devices and their sensors, and *external resources*, like storage, are homogeneously managed. Thanks to the API provided by WEAR-IT, the MoveUp wearable expert system can access data from sensors on wearable devices in a transparent way.

The *Domain Dependence* label in Figure 6 illustrates the main technologies and operating systems currently available: wearable expert systems obtain data in a format depending on the desired sensor and wearable device. This format varies according to the domain: sensors available in Android Wear are different from sensors available on a BLE device. The WEAR-IT API makes the WES obtain data from the sensor without being aware of the sensor features. Solid arrows in Figure 6 show the current development of WEAR-IT API from the domain dependence perspective; dashed arrows specify under-development or possible future implementations.

Figure 7 presents a typical situation where wearable environments are exploited. Each user has some applications installed on his/her smartphone. Moreover, the smartphone is equipped with the WEAR-IT framework. The applications interact with WEAR-IT to acquire data from wearable devices distributed around the environment. Storage of data is on the cloud, to maximize the sharing of data among different users and/or applications, if involved in the same overall project. To this aim, WEAR-IT can communicate with opportune IoT platforms such as KAA (https://www.kaaproject.org/).

The *MoveUp* app supports personalized training programs in people at risk from the cardiovascular disease point of view. Physical activity (PA) is a very important factor to prevent chronic diseases; although guidelines have been produced about the amount of PA to accomplish, and they are continuously updated, much of the population continues to follow a sedentary lifestyle [40]. Traditional behavioral interventions have produced scarce results from the PA promotion point of view [41]; on the contrary, the availability of new, wearable technologies has allowed developing new applications to support people in modifying their behavior with respect to PA.

In the case study, the user exploits an application suggesting him/her how much physical activity to accomplish during the week; the evaluation is based both on quantitative, physical (e.g., “how much time do you train during a week?”) and qualitative, psychological variables (e.g., “how well did you feel during the training session?”). From the physical point of view, the *metabolic equivalent of task* (MET) variable is used to estimate the amount and intensity of PA accomplished. The approach is based on guidelines provided by WHO [42] and it has been validated during an experiment involving 60 people, as presented in [43]. Here, the focus is not on the application itself, but on how the WEAR-IT introduction can be exploited to increase efficiency in the design and implementation of mHealth applications, as well as thinking at their development as an iterative process where new modules can be added that use the same wearable devices and data.

To develop the MoveUp reasoning strategy, the relationships between MET and *heart rate* (HR) has been used, given by the following formula [44]:(1)$$MET=4*Time^{MPA}+8*Time^{IPA}$$
where $Time^{MPA}$ and $Time^{IPA}$ are the periods of time the subject is involved in *moderate* and *intense* physical activity, respectively, measured in minutes.

Thus, the wearable environment in Figure 7 is configured to detect HR values from one connected wearable device starting at a given *time instant* and *frequency* and for a given *period of time*, as suggested by the application. Let us suppose that the user must accomplish 120 MET of physical activity: according to WHO guidelines (see https://www.who.int/dietphysicalactivity/factsheet_recommendations/en/), 120 METs correspond to 30 min of MPA or 15 min of IPA. A PA session is defined *moderate* if the registered HR values are in the range $\left[\frac{6*MHR}{10},\frac{7*MHR}{10}\right]$, with MHR=220−age is the subject’s *maximum heart rate*, depending on his/her *age*. A PA session is defined *intense* if the registered HR values are in the range $\left(\frac{7*MHR}{10},\frac{8*MHR}{10}\right]$.

This means that HR values should be sampled from a HR sensor over a 30-min time horizon. Algorithm 3 shows a sketch of Algorithm 1 related to the HR detection in MoveUp. The wearable device used in the example is a $PulseOn^{TM}$ (see https://pulseon.com/) smartwatch.

Figure 8 shows how WEAR-IT is involved in the development of the related wearable expert system. Starting from raw data acquisition from wearable devices (e.g., the heart rate), the MoveUp module provides suggestions about the amount of physical activity to accomplish week by week (further details about the conceptual model in [45]), thanks to WEAR-IT API use.
   **Algorithm 3** WEAR-IT API use in MoveUp.**Require:**sp**Ensure:**$s,wd,data_{s}$     Comment: scan the wearable environment     $\left\{PulseOn^{TM},\ldots \right\}=scan\left(sp\right),i\in \left[1\ldots k\right]$     Comment: connect to $PulseOn^{TM}$ suitable for the application     $connect(PulseOn^{TM},sp)$     Comment: identify the sensors in the $PulseOn^{TM}$ wearable device     $\left\{s_{PulseOn^{TM}}^{j}\right\}=classify\left(sensors\left(PulseOn^{TM}\right)\right),j\in \left[1\ldots m\right]$     Comment: select the needed sensor of type HEART_RATE     $s=select(\left\{s_{PulseOn^{TM}}^{j}\right\},HEART\_RATE)$     Comment: measurement takes place for 30 min at default frequency; stop is not mandatory     $play(s,t_{0},t_{0}+30\;\min)$     Comment: data gathered by HR sensor are stored in XLS format     **return**
$store(s,PulseOn^{TM},Path_{data_{s}},data_{s},XLS)$

## 5. Discussion

In this paper, a conceptual and computational approach to the design of IoT-based intelligent systems is presented. The main idea is the definition of a wearable environment where intelligent systems can interact with sensors in order to maximize the performance of their reasoning strategy. To do this, a *middleware* layer has been inserted between the application and data layers, to provide the former with an opportune API to gather data from the latter. The main benefits emerging from this choice, represented in the diagram of Figure 6, are *interoperability* among applications at reasoning level and *reliability* of data made available to applications. In the following, these two aspects are further explained.

### 5.1. Interoperability at Reasoning Level

It is important to notice that such API has been designed and implemented taking care of the goal of a wearable environment: to define a mean to allow an application at reasoning level to perform its decision making process being sure to find and obtain necessary data.

Figure 9 refers to the case study above: two distinct BLE devices, namely $PulseOn^{TM}$ (see https://pulseon.com/) and $NokiaSteelHR^{TM}$ (see https://www.withings.com/uk/en/steel-hr) smartwatches, are included in the wearable environment definition. Both of them have a HR sensor, thus the MoveUp application could choose the one or the other indifferently, from its performance point of view. Unfortunately, if the $NokiaSteelHR^{TM}$ were selected to query its sensors, the WEAR-IT framework would not be able to discover available data sources. This means that BLE configuration of the device is not standard, thus no data can be delivered to a WES.

This is a problem common to many wearable devices, for which opportune APIs are (maybe partially) not available and they can only be interrogated through their official apps (e.g., $HealthMate^{TM}$ (see https://www.withings.com/nl/en/health-mate)); the possibility to connect a proprietary app such as Health Mate to a middleware such as WEAR-IT, in order to overcome the problem described so far could open new possibilities of developing innovative application in mobile domains; one possible solution, according to the wearable environment definition, would be to extend the APIs set of WEAR-IT to allow an application $a_{i}$ “playing” another application $a_{j}$:$$API_{a_{i_{a_{j}}}}=\left\{play\left(a_{j}\right),stop\left(a_{j}\right),store\left(a_{j}\right)\right\}$$

An example of such interoperability is the development of MoveUp VR.The users of the MoveUp module are people characterized by sedentary lifestyle and great difficulties to perform regular physical activities due to many reasons. They result to be very frail psychologically and they could be discouraged by well-known barriers [46].

To support the users to plan their training session, an interactive function for calculating the best route to run across has been designed and implemented. Figure 10 shows the information flow concerning this goal. First, the user chooses the starting point on the map: this point could be concrete (e.g., the address of the hotel where the person stays during a business trip) or desirable (e.g., a beautiful point of interest). Then, an input form is activated where the following data must be inserted:*Duration of the PA session*: This value can be calculated from the MET amount suggested by MoveUp (remind that 120 MET are equivalent to 30 min of moderate PA). The user is free to setup the value according to his/her wishes.*Speed*: This value is set by default at 6 km/h, but the user can modify it according to his/her conditions.*Direction*: This value is set by default at 0 degrees, but the user can modify it exploiting the *rotation vector sensor* (if available) provided by WEAR-IT; for example, he/she could point towards a POI he/she sees on the map, in order to force the application to consider it in the calculation.

At the end of the calculation, the proposed route is highlighted on the map and the user can simulate to run it through the $StreetView^{TM}$ activation. Doing so, he/she will be able to decide if the route neighborhood fits with his/her expectations or modifying it otherwise. In this case, the MET value exploited in the route calculus is asked to the MoveUp application; the MET value is not gathered from a sensor, being derived from HR thanks to the MoveUp decision’s model. Much work must be done to extend the WEAR-IT API to enable an application to interface with another one to obtain data, and this is being addressed by ongoing work.

### 5.2. Data Reliability

Expert systems, and wearable expert systems too, are generally able to work if some data are missing. The availability of faulty data is more critical, since the decision making process could be erroneously conducted. The probability that measures from wearable devices are incorrect is high, especially when data gathered concern medical parameters such as HR [47]. One of the most important goals of the wearable environment described is allowing applications at reasoning level to deal with reliable data. To this aim, some experiments on the wearable environment storage have been conducted.

Given an IoT device to measure *temperature*, *humidity*, *luminosity* and *battery voltage*, data gathered have been archived in the KAA platform for a period of 36 days. The resulting table contained 2,313,682 records. Table 13 summarizes the number of anomalies detected on them.

Figure 11 shows the distribution of values for the temperature and humidity variables. It is possible to notice how the temperature curve is uniformly distributed between 10 and 35 ${}^{\circ }$C; these values are much more than others, constituting the set of correct values for the temperature detection. Moreover, there are some peaks of anomaly values that are very frequent. i.e., 0.51 and 121.49 ${}^{\circ }$C. About humidity, the distribution is Gaussian between 24% and 60% and uniform between 0% and 24%; moreover, there are very few measurements in the (60%, 100%] range, with a significant peak of anomaly at 114.45% (about 4000 units).

Figure 12 presents the distribution of values for the luminosity and voltage parameters. The luminosity distribution is linear on the whole interval, except for peaks at the limits (0 and 1600 lux respectively) and some anomalies at 41, 43 and 48 lux values. Finally, the battery voltage distribution is Gaussian, without any significant peak or anomaly.

The analysis of curves of parameters values allowed deriving opportune sets of acceptable values for sensors measurements, as shown in Table 14. The temperature range was extended to [−15 ${}^{\circ }$C, 50 ${}^{\circ }$C] to take care of winter and summer periods (the experiment was conducted łlast spring).

The last step of the intervention is the nullification of out of bounds values. The consequent loss of information due to the deletion of data from the storage is compensated by the lower probability of taking wrong decisions as a consequence of sensors failures. Figure 13 shows the decrease of standard deviation of variables at the end of this intervention. According to the knowledge domain, out of bound values could be substituted by reasonable, standard values: for example, in the MoveUp case study, a possible strategy to improve the data quality could be replacing extremely high outlier values of HR with user’s MHR.

## 6. Conclusions

In this paper, we propose the *wearable environment* notion and architecture as a means to cast into a unique conceptual and computational framework data acquisition from sensors, data representation through wearable devices and their use by means of wearable expert systems. The heart of this notion is the data acquisition and representation layer, where the WEAR-IT platform has been designed and implemented to build an efficient, reliable and scalable bridge between raw data and applications.

To our knowledge, only three applications provide services comparable to WEAR-IT: *SensorCap, Sensor Toolbox and LightBlue Explorer* (see https://play.google.com/store/apps/details?id=br.ufmg.dcc.ssig.sensorcap; https://play.google.com/store/apps/details?id=com.galaxy.sensortoolbox; and https://play.google.com/store/apps/details?id=com.punchthrough.lightblueexplorer). With respect to them, only WEAR-IT can be integrated (see Figure 8) into a wearable environment at the moment, while the others can be used as external services to collect and store data that can be then elaborated by the application.

Indeed, data reliability is one of the most important factor for WES development. Consequently, given that, in a wearable environment, WEAR-IT acts as an intermediate data collector, it is important to ensure that the *quality* of data stored is high enough (see Section 5.2). This issue is currently addressing our research [48].

Finally, a future extension of the present work concerns the definition of an extended wearable environment infrastructure, where many users, each one equipped with a *personal wearable environment*, could be asked to join an *integrated*, *geographically distributed* and *scalable* wearable environment. In this environment, an application could query end-points and acquire data from them according to their position and reachability from mobile nodes [33].

## Figures and Tables

**Figure 1 sensors-20-05970-f001:**
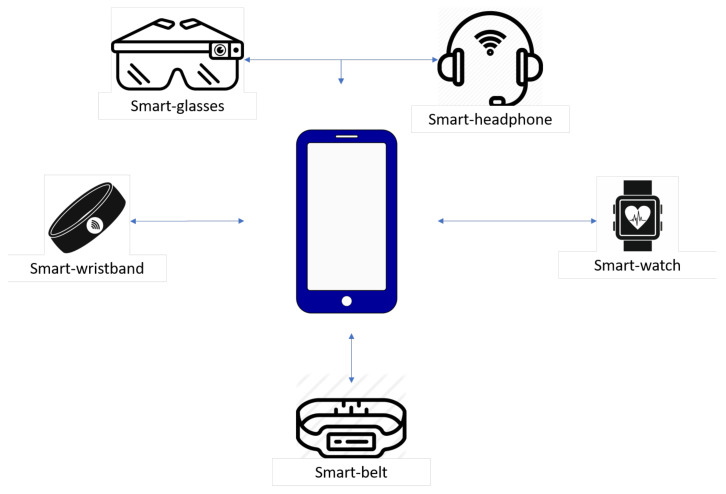
A sketch of the wearable devices involved in the definition of a wearable environment.

**Figure 2 sensors-20-05970-f002:**
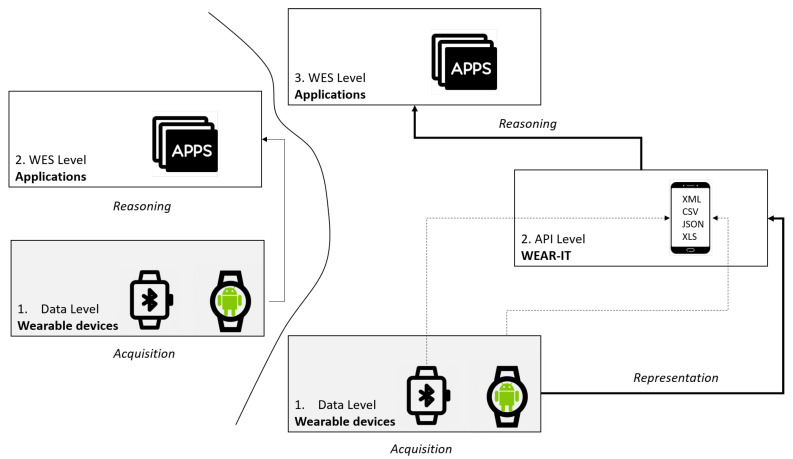
The API Level introduction: from a two-tier to a three-tier architecture.

**Figure 3 sensors-20-05970-f003:**
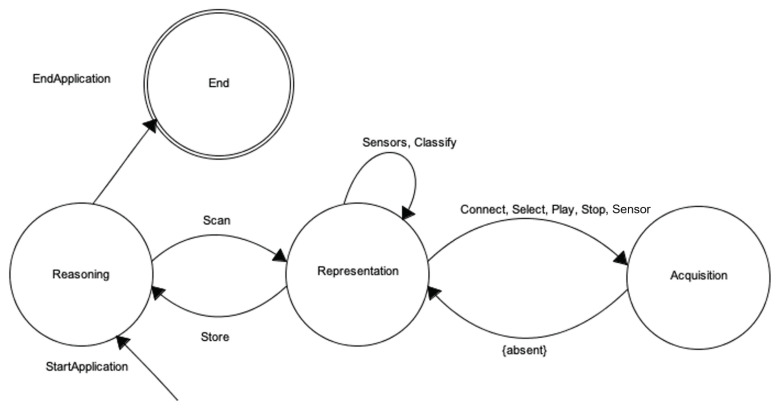
A Wearable Environment representation as a finite state machine.

**Figure 4 sensors-20-05970-f004:**
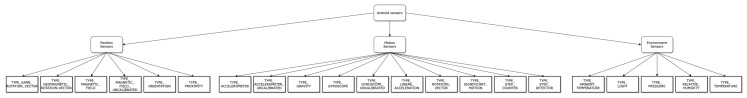
Hierarchical representation of sensors in $Android^{TM}$.

**Figure 5 sensors-20-05970-f005:**
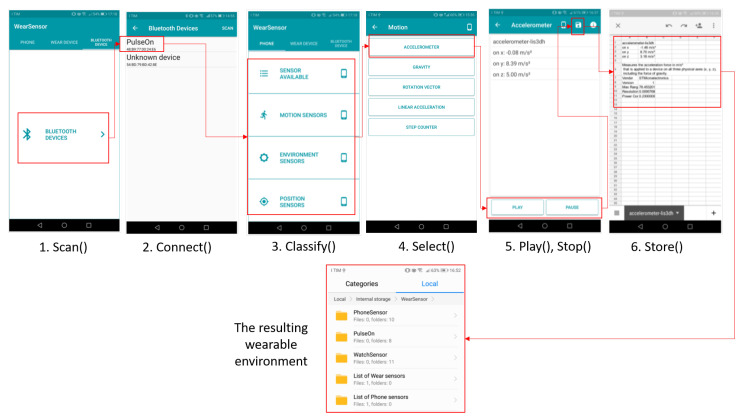
An application of Algorithm 1 for the generation of a wearable environment.

**Figure 6 sensors-20-05970-f006:**
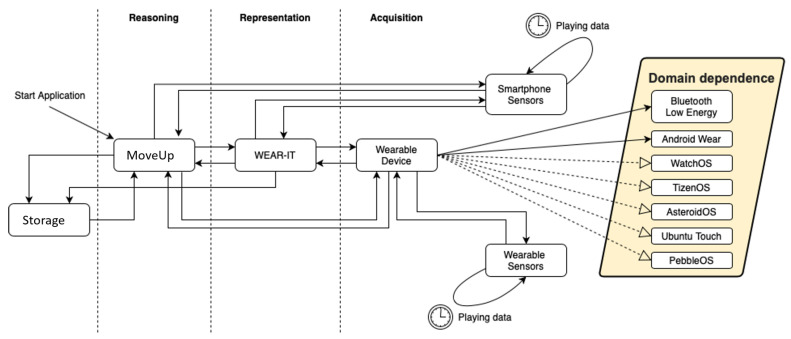
The wearable environment FSM model implemented into a case study.

**Figure 7 sensors-20-05970-f007:**
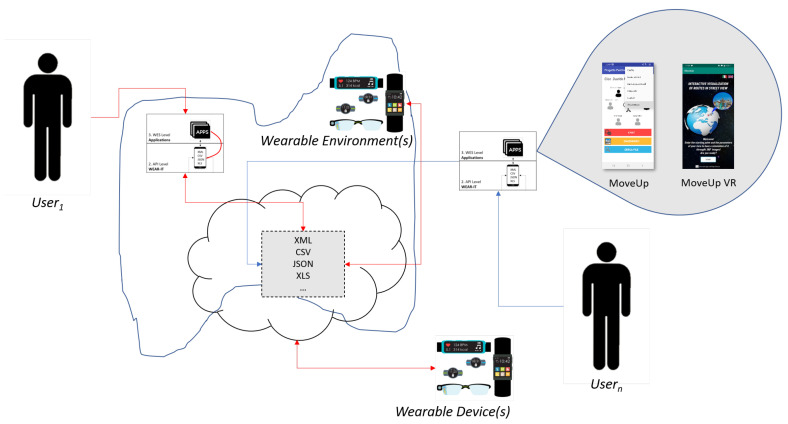
Information flow in the proposed scenario.

**Figure 8 sensors-20-05970-f008:**
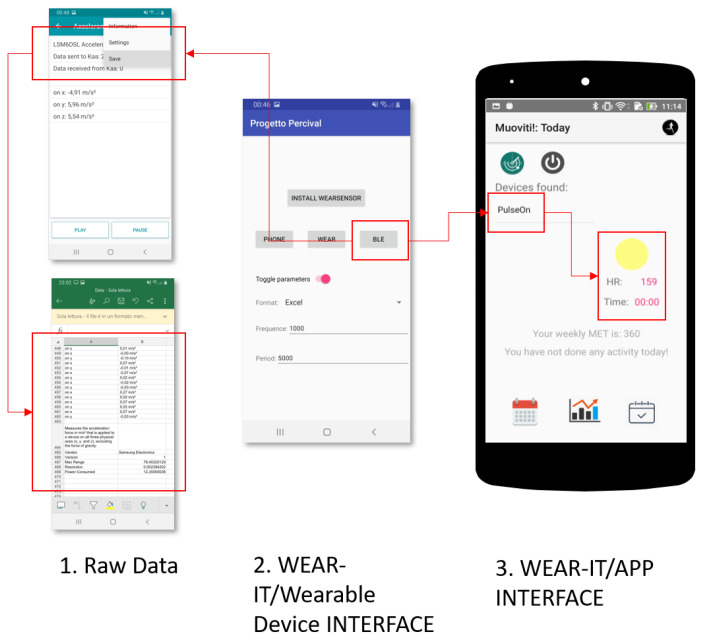
The three stages of app development in the case study.

**Figure 9 sensors-20-05970-f009:**
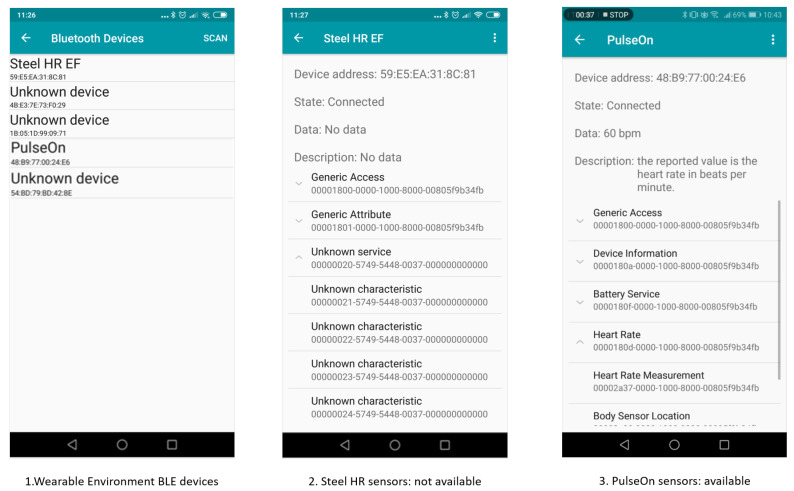
BLE devices comparison in WEAR-IT.

**Figure 10 sensors-20-05970-f010:**
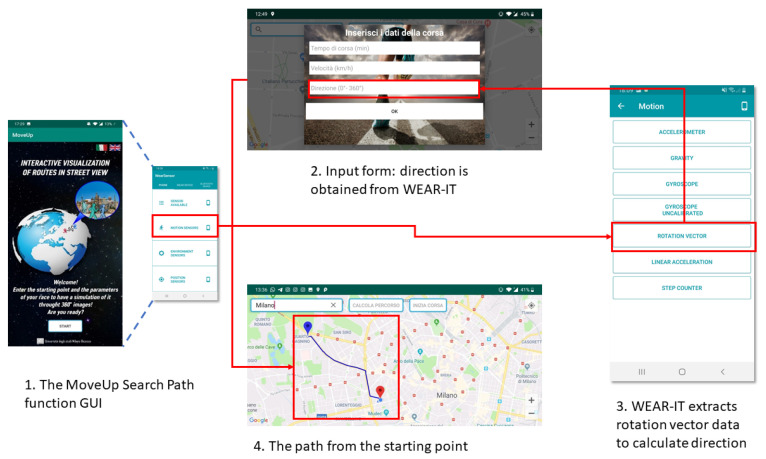
The information flow to calculate the path for running in MoveUp.

**Figure 11 sensors-20-05970-f011:**
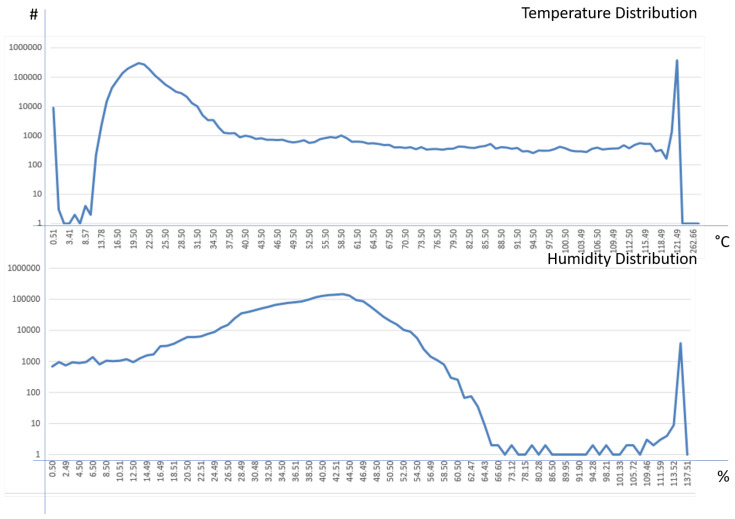
Distribution of luminosity and battery voltage values archived in the KAA storage.

**Figure 12 sensors-20-05970-f012:**
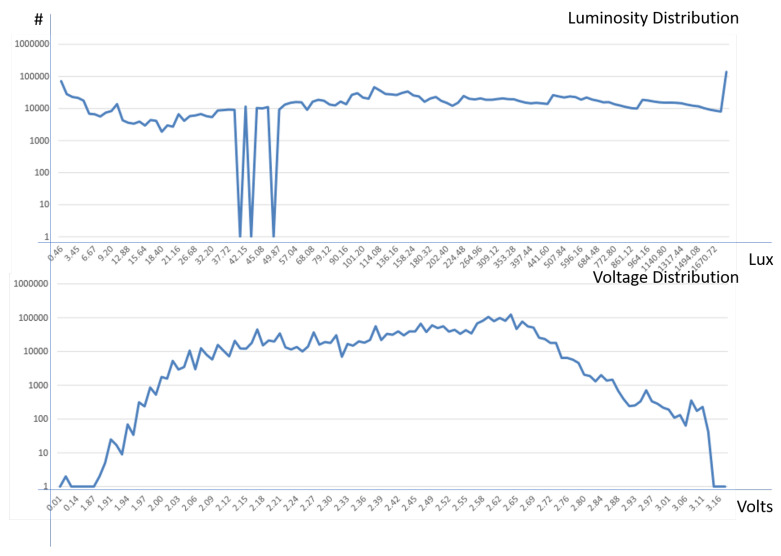
Distribution of temperature and humidity values archived in the KAA storage.

**Figure 13 sensors-20-05970-f013:**
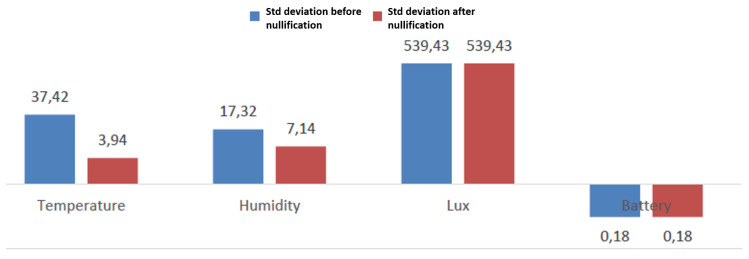
Standard deviation of temperature, humidity, luminosity and voltage values before and after nullification.

**Table 1 sensors-20-05970-t001:** State transition table of a wearable environment.

	Input	Scan	Connect	Sensors	Classify	Select	Play	Stop	Store	*{Absent}*
States										
⟶**Reasoning**		Representation	⌀	⌀	⌀	⌀	⌀	⌀	⌀	⌀
**Representation**		⌀	Acquisition	Representation	Representation	Acquisition	Acquisition	Acquisition	Reasoning	⌀
**Acquisition**		⌀	⌀	⌀	⌀	⌀	⌀	⌀	⌀	Representation

**Table 2 sensors-20-05970-t002:** Output function for each state of a wearable environment.

Reasoning State	
g(S×I)	U
g (Reasoning, Scan)	List <Wearable Device >
**Representation State**	
g(S×I)	U
g (Representation, Connect)	Wearable device
g (Representation, Sensors)	*⌀*, List <Sensor >
g (Representation, Classify)	Tree <Sensor>
g (Representation, Select)	Sensor
g (Representation, Play)	True/False
g (Representation, Stop)	True/False
g (Representation, Store)	Path
**Acquisition State**	
g(S×I)	U
g (Acquisition, Wearable device)	True/False
g (Acquisition, Sensor)	True/False
g (Acquisition, True/False)	List<SensorValue>

**Table 3 sensors-20-05970-t003:** Definition of a primitive structure.

Return Value	Method Name, Parameters, Description
Value	name Primitive(parameter 1, parameter 2, ..., parameter n) Description

**Table 4 sensors-20-05970-t004:** *The Scan*() definition.

Return Value	Method Name, Parameters, Description
List <WearableDevice>	*Scan*() Returns the collection of wearable devices available in a wearable environment.

**Table 5 sensors-20-05970-t005:** The *Connect(sp, wd)* definition.

Return Value	Method Name, Parameters, Description
True/False	Connect (Smartphone sp, WearableDevice wd) Returns *True* if pairing sp and wd succeed; *False* otherwise.

**Table 6 sensors-20-05970-t006:** The *Sensors(sp)* and *Sensors(wd)* definition.

Return Value	Method Name, Parameters, Description
List <Sensor>	Sensors (*Smartphone sp*) Returns the list of sensors on the smartphone on the smartphone *sp*.
List <Sensor>	Sensors (*WearableDevice wd*) Returns the list of sensors on the wearable device *wd*.

**Table 7 sensors-20-05970-t007:** The *Classify(sp)* and *Classify(wd)* definition.

Return Value	Method Name, Parameters, Description
Tree <Sensor>	Classify (*Smartphone sp*) Returns a tree-like cluster of sensors available on the smartphone *sp*.
Tree <Sensor>	Classify (*WearableDevice wd*) Returns a tree-like cluster of sensors available on the wearable device *wd*.

**Table 8 sensors-20-05970-t008:** The *Sensor(sp, s)* and *Sensor(wd, s)* definition.

Return Value	Method Name, Parameters, Description
True/False	Sensor (Smartphone sp, Sensor s) Returns True if the sensor *s* is available on the smartphone *sp*; False otherwise.
True/False	Sensor (WearableDevice wd, Sensor s) Returns True if the sensor *s* is available on the wearable device *wd*; False otherwise.

**Table 9 sensors-20-05970-t009:** The *Select(sp, s)* and *Select(wd, s)* definition.

Return Value	Method Name, Parameters, Description
Sensor	Select (Smartphone sp, Sensor s) Returns the ID code of the sensor *s* if it has been correctly activated on the smartphone sp; an empty string otherwise
Sensor	Select (WearableDevice wd, Sensor s) Returns the ID code of the sensor *s* if it has been correctly activated on the wearable device wd; an empty string otherwise.

**Table 10 sensors-20-05970-t010:** The Play(sp, s, t0, tn, f) and Play(wd, s, t0, tn, f) definition.

Return Value	Method Name, Parameters, Description
List <SensorValue>	Play (Smartphone sp, Sensor s, Time $t_{0}$, Time $t_{n}$, Frequency f) Returns a list of values gathered by sensor *s* on smartphone sp according to the parameters settings.
List <SensorValue>	Play (WearableDevice wd, Sensor s, Time $t_{0}$, Time $t_{n}$, Frequency f) Returns a list of values gathered by sensor *s* on wearable device wd according to the parameters settings.

**Table 11 sensors-20-05970-t011:** The Stop(sp, s) and Stop(wd, s) definition.

Return Value	Method Name, Parameters, Description
Void	Stop (Smartphone sp, Sensor s) Data detection on sensor *s* of smartphone sp is terminated.
Void	Stop (WearableDevice wd, Sensor s) Data detection on sensor *s* of wearable device *wd* is terminated.

**Table 12 sensors-20-05970-t012:** The *Store*() definition for smartphones and wearable devices.

Return Value	Method Name, Parameters, Description
Path	Store (Sensor s, Smartphone sp, Artifact a, List<SensorValue> sensorValues, Format f) Stores a set of data *sensorValues* gathered by sensor *s* within the *Artifact a*, according to the *format* *f*. In case of success, the *path* to the storage is returned; an error code is generated otherwise.
Path	Store (Sensor s, Wearable device wd, Artifact a, List<SensorValue> sensorValues, Format f) Stores a set of data *sensorValues* gathered by sensor *s* within the *Artifact a*, according to the *format* *f*. In case of success, the *path* to the storage is returned; an error code is generated otherwise.

**Table 13 sensors-20-05970-t013:** Analysis of anomalies on data measurements.

Values	Parameter			
	Temperature	Humidity	Luminosity	Voltage
Null	901 (0.03%)	902 (0.03%)	93,880 (4%)	526 (0.02%)
Outliers	383,443 (16.6%)	299,084 (12.9%)	0	8 (0.03%)

**Table 14 sensors-20-05970-t014:** Acceptable values for temperature, humidity, luminosity and voltage.

Parameter	Measurement	
	Minimum Value	Maximum Value
Temperature	−15 ${}^{\circ }$C	50 ${}^{\circ }$C
Humidity	0% h	100% h
Luminosity	0 Lux	2500 Lux
Voltage	1, 5 Volts	3, 5 Volts

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
