# Peer review of "An API for Wearable Environments Development and Its Application to mHealth Field"

_sensors, 2020, doi:10.3390/s20215970_

Round 1

Reviewer 1 Report

A use case og the proposed API could be helpful for the reader.

The authors should provide a more extended comparison of the proposed API with similar frameworks.

Author Response

Dear reviewer,

thank you very much for your valuable feedback.

I have modified the paper, adding an example section at the end of the theoretical section. The section is number 3.3.

About the comparison of the API with similar frameworks, I've motivated the differences from others in the Discussion section. I left the comparison with similar frameworks from the computational perspective.

I hope this revision will meet your requests.

Reviewer 2 Report

In this work the authors present systems able to connect real-world devices in order to build effective mobile applications. This works is based in previous work they presented introducing a wearable expert systems as a  paradigm to develop knowledge based systems, focusing on the conceptual model necessary to develop a complete rule-based system from a bottom-up analysis of the relations among the data collected in the environment.

This paper is interesting and of good technical depth, the architecture and the rationale are rather well explained but here are some points that to be clarified (week points of this paper)

  • How is the proposed system able to improve interoperability and standardization in detecting and storing acquired data - and more importantly medical data. It is well known that many standards exist and medical device providers (extending connectivity protocols i.e. BLE) are able to provide. For example we provide some input of the:
  • Health Level Seven International - Homepage | HL7 hl7.org
  • FHIR v4.0.1 - HL7.org hl7.org https://hl7.org/FHIR/
  • There is a section about BLE that is not clear if it is wanted or the reason (see comment about sensors - most of them exist on a single device)  
  • The paper discusses wearables but the provided Table 2 discusses about devices that exist usually on mobile phone and/or wristbands, and this is against Figure 1 paradigm that clearly mentions a plethora of devices. The question is thus how all devices are categorized and what kind of data are provided. The are plenty of interesting works out there identifying ways to gather sparce data of low requirement sin bandwidth – how they differ from this work
  • The authors discuss of an API of commercial based sensors that a) is clear that major manufacturing companies have already identified such an API (i.e. APPLE, SAMSUNG) and allow you on a need to be basis to access data/metatadata – agai how this work is of difference
  • An interesting topic also would be to emphasize on the fusion of information but then the authors need to reference related material i.e. how to use ehealth through smart enviroments: Spanakis, E.G., Lelis, P., Chiarugi, F., & Chronaki, C. (2006). R&D challenges in developing an ambient intelligence eHealth platform. EMBEC 2005, Prague, Czech Republic, 2005, 20-25 Nov. (vol. 11 pp.1727-1983), this is an early example of refs in this topic
  • The goals presented and the respected formulas are they validated? Are they part of a protocol of some sort? Looking at EMBC conference I found this : Congestive Heart Failure Risk Assessment Monitoring through Internet of things and mobile Personal Health Systems. 40th Annual International Conference of the IEEE Engineering in Medicine and Biology Society (EMBC), 18-21 July 2018, Honolulu, HI, USA, 29 October 2018. IEEE (1557-170X/1558- 4615/https://doi.org/10.1109/EMBC.2018.8513024 ), doi: 10.1109/EMBC.2018.8513024. This is about using sensing devices and rule based IoT in order to manage heatlh – please ref, and extent related work of this topic.

Author Response

Reviewer 2

In this work the authors present systems able to connect real-world devices in order to build effective mobile applications. This works is based in previous work they presented introducing a wearable expert systems as a  paradigm to develop knowledge based systems, focusing on the conceptual model necessary to develop a complete rule-based system from a bottom-up analysis of the relations among the data collected in the environment.

This paper is interesting and of good technical depth, the architecture and the rationale are rather well explained but here are some points that to be clarified (week points of this paper)

Dear reviewer,

Thank you very much for your valuable feedback.

How is the proposed system able to improve interoperability and standardization in detecting and storing acquired data - and more importantly medical data. It is well known that many standards exist and medical device providers (extending connectivity protocols i.e. BLE) are able to provide. For example we provide some input of the:

Health Level Seven International - Homepage | HL7 hl7.org

FHIR v4.0.1 - HL7.org hl7.org https://hl7.org/FHIR/

I completely revised the paper to explain that the focus here was not on the medical treatment of patients in the mHealth field, but to propose a framework to allow the development of applications in that field. Section 3 has been completely rewritten to this aim, focusing on the definition of a wearable environment as a FSM, with a complete operational semantics. The MoveUp module description has been reduced and centered on the API use to obtain data from sensors (HBR in particular) according to application needs. The aim of the Wear-It API is not to produce a new standard at the moment, especially in the mHealth sector.

There is a section about BLE that is not clear if it is wanted or the reason (see comment about sensors - most of them exist on a single device) 

The BLE section has been removed since it was not useful anymore.

The paper discusses wearables but the provided Table 2 discusses about devices that exist usually on mobile phone and/or wristbands, and this is against Figure 1 paradigm that clearly mentions a plethora of devices. The question is thus how all devices are categorized and what kind of data are provided. The are plenty of interesting works out there identifying ways to gather sparce data of low requirement sin bandwidth – how they differ from this work

Table 2 does not exist anymore. The paper content has been completely revised to make clearer the aims of the Wear-It framework, as well as its API. The comment is really interesting and I will consider it for future works. Thank you.

The authors discuss of an API of commercial based sensors that a) is clear that major manufacturing companies have already identified such an API (i.e. APPLE, SAMSUNG) and allow you on a need to be basis to access data/metatadata – agai how this work is of difference

I briefly discuss about the need for an API capable to query sensors mounted on many different devices. The main problem I found with existing commercial APIs is that they don’t allow a complete work on wearable devices, since they are partially or totally proprietary. In the Percival project we tested many devices for HBR detection: in the paper I propose a comparison between a PulseOn and a Nokia Steel HR smartwatches: while the PulseOn sensors are readable, the Nokia Steel HR ones are not. Thus, the usage of the proprietary app is necessary. This means that an application at reasoning level should not be able to complete its own decision making process, unless it finds a suitable HBR sensor around, being free to chose the best one according to its wishes. This is the reason why Wear-It framework and the Wearable Environment notion have been defined.    

An interesting topic also would be to emphasize on the fusion of information but then the authors need to reference related material i.e. how to use ehealth through smart enviroments: Spanakis, E.G., Lelis, P., Chiarugi, F., & Chronaki, C. (2006). R&D challenges in developing an ambient intelligence eHealth platform. EMBEC 2005, Prague, Czech Republic, 2005, 20-25 Nov. (vol. 11 pp.1727-1983), this is an early example of refs in this topic

Thank you for this comment, I was not able to consider this topic due to the huge amount of work to revise the paper. I will consider it for future works.

The goals presented and the respected formulas are they validated? Are they part of a protocol of some sort? Looking at EMBC conference I found this : Congestive Heart Failure Risk Assessment Monitoring through Internet of things and mobile Personal Health Systems. 40th Annual International Conference of the IEEE Engineering in Medicine and Biology Society (EMBC), 18-21 July 2018, Honolulu, HI, USA, 29 October 2018. IEEE (1557-170X/1558- 4615/https://doi.org/10.1109/EMBC.2018.8513024 ), doi: 10.1109/EMBC.2018.8513024. This is about using sensing devices and rule based IoT in order to manage heatlh – please ref, and extent related work of this topic.

The mHealth domain has been considered as a case study to show the applicability of the Wear-It API. I reduced the MoveUp description for sake of clarity. Anyway, I put a reference in the paper to give more information about the validation of the MoveUp approach according to experts’ knowledge (i.e. psychologists involved in Behavior Chang Interventions).

Reviewer 3 Report

This paper presents some details and scenarios of a sensor-based environment for mHealth applications, including those based on virtual reality. The implementation is based on other frameworks, including the author's previous work; a comparison with other frameworks is provided, but appears somewhat superficial.

The paper lacks a clear structure, and it is quite unclear what the objective of the paper is. There are quite many subjects presented (such as an API, kind of framework, some algorithms, some apps including screenshots, virtual reality-based applications). Evaluation criteria and research questions are not given. The conclusion is rather vague and not really based on the presented matter.

Regarding the structure, Section 3 should be named "Architecture" (or similar; not "Results"). Section 3 contains an unstructured collection of architecture elements, algorithms, apps and frameworks, screenshots, evaluations, etc.

It is unclear what the role of the presented apps is. Further, the role of virtual reality is unclear. It seems that virtual reality is an additional element that is not necessary for the architecture or API as such. Further, virtual reality seems not being relevant for the API.

The entire article is based on many product names, project names, framework names; thus, a lack of generality might be present. Some of these names are introduced without any explanation why these are relevant; e.g., Kaa, WEAR-IT, etc. These platforms relevant for the framework should be introduced earlier in the paper.

It seems that the author describes implementation work. For some parts it is not made obvious why some of the design decisions were made in just that way.

The API-framework seems to be rather standard. The API is not compared with other APIs or architectures. There are architectures for sensors, where many of the sensor- and synchronisation details are hidden, for instance in message oriented architectures, semantic web-based architectures, etc. Such architectures also have security built-in, which seems not the case with the author's implementation.

As a framework, it seems that the API is rather bound to current technologies (such as BLE).

The apps are inspired from mHealth applications. However, the application presented in Section 3.2.2 seems to follow a simplified technical approach instead of an approved/evidence-based healthcare/medical approach. Note that in health applications (beyond usual training apps) there are medical protocols or pathways that need to be followed. Note that wrong recommendations from an app potentially could harm the patient. It is unclear whether the procedure described from line 353 is medically sound. Please discuss this matter with scientists from the nursery/medical/healthcare domain.

Minor comments:

list of abbreviations line 524 is very useful. Please sort the entries in this table alphabetically.

In the abstract you mention that smartphones are equipped with many sensors, hereunder thermometers. Currently, I do not know of smartphones that are equipped with a thermometer sensor where the values can be used in apps. Thermometers might be available as external devices, but usually not built-in.

paragraph starting line 60 is already a kind of conclusion in the introduction section. Please move to conclusion or similar.

line 69; style: "A massive work" --> "Much work"

line 70, style: avoid lengthy sentence in front of the references 5 and 6. why are these surveys well-known?

Please avoid references as nouns; e.g., ref 23, etc. Further, remove "in" in front of references, e.g., line 107: "proposed [29]"; line 110: "Mobility in WSN is .... reasons [30] ..." and so on

line 185: Table with capital T

Table 1: the API seems rather small and straightforward. How does the API compare with other APIs?

Figure 3 is unclear.

line 289: why alphabetically ordered?

line 302: Part with capital P

Figure 5: much space for little information ...

line 321: what do you mean by "profitably exploited" ? how is this measured?

line 325: typo: word -> world

line 386: why is this UI sophisticated?

The role of Section 4.1 and the scenario presented are unclear. How does this fit with the discussions around the API?

line 411: The role of virtual reality is unclear. Why are all these products/trademarks mentioned? This part reads more like a white-paper than a scientific article.

line 434: references to SensorCap, Sensor Toolbox, LightBlue are not given.

line 452: Trough -> Through

line 466: unclear; what do you mean by "fully integrated"?

lines 478ff: references not given: PulseOn, NokiaSteelHR

line 489: strange sentence: why does an activity stem from a fact?

Author Response

Reviewer 3

This paper presents some details and scenarios of a sensor-based environment for mHealth applications, including those based on virtual reality. The implementation is based on other frameworks, including the author's previous work; a comparison with other frameworks is provided, but appears somewhat superficial.

Dear reviewer,

Thank you for valuale feedback. I hope the proposed revision will meet your requests.

The paper lacks a clear structure, and it is quite unclear what the objective of the paper is. There are quite many subjects presented (such as an API, kind of framework, some algorithms, some apps including screenshots, virtual reality-based applications). Evaluation criteria and research questions are not given. The conclusion is rather vague and not really based on the presented matter.

The paper content has been completely revised in order to make clearer the scope of the wearable environment notion and the Wear-It framework within it.

Regarding the structure, Section 3 should be named "Architecture" (or similar; not "Results"). Section 3 contains an unstructured collection of architecture elements, algorithms, apps and frameworks, screenshots, evaluations, etc.

I changed the paper organization according to your suggestion: now, Section 3 has been devoted to collect all the theoretical and practical aspects of the Wear-It definition.

It is unclear what the role of the presented apps is. Further, the role of virtual reality is unclear. It seems that virtual reality is an additional element that is not necessary for the architecture or API as such. Further, virtual reality seems not being relevant for the API.

The MoveUp role has been limited to a Wear-It case study in Section 4, to make clearer that the paper focus is on the architectural patterns to allow applications to gather data from sensors. The MoveUp VR description has been reduced and moved to discussion section as an example of interoperability among apps.

The entire article is based on many product names, project names, framework names; thus, a lack of generality might be present. Some of these names are introduced without any explanation why these are relevant; e.g., Kaa, WEAR-IT, etc. These platforms relevant for the framework should be introduced earlier in the paper.

The paper content has been completely revised. I hope their role in the wearable environment deifinition is clearer now. In particular, the references to KAA have been removed, thus they are not useful anymore. Wear-It is the subject of Section 3, being the focus of the paper.

It seems that the author describes implementation work. For some parts it is not made obvious why some of the design decisions were made in just that way.

I removed all the impleenation parts that were not clear. In particular, table about AndroidWear and BLE, as well as implementation details about ATT and GATT profiles havde been removed; instead, I have further developed the conceptual description of the representation level of a wearable environment, presenting it as a FSM with an opportune operational semantics. See Section 3 for all details about that.

The API-framework seems to be rather standard. The API is not compared with other APIs or architectures. There are architectures for sensors, where many of the sensor- and synchronisation details are hidden, for instance in message oriented architectures, semantic web-based architectures, etc. Such architectures also have security built-in, which seems not the case with the author's implementation.

I discussed this point in Section 5. The paper focus is not the comparison with existing standards and frameworks, but the development of an API to enable applications at reasoning level to complete their decision making process being sure to have necessary data at the right moment. For this reason, I didn’t compare Wear-It with existing standards, but with similar applications, that is more appropriate in my opinion, according to the paper scope. 

As a framework, it seems that the API is rather bound to current technologies (such as BLE).

That’s right. I focused on Android Wear and BLE at the moment, since they are the most used protocols for the applications we are developing. Anyway, we are working to extend the wearable environment notion to different technologies and approaches, including network slicing solutions too. I will discuss this point at the WiMob 2020 Conference, next October.  

The apps are inspired from mHealth applications. However, the application presented in Section 3.2.2 seems to follow a simplified technical approach instead of an approved/evidence-based healthcare/medical approach. Note that in health applications (beyond usual training apps) there are medical protocols or pathways that need to be followed. Note that wrong recommendations from an app potentially could harm the patient. It is unclear whether the procedure described from line 353 is medically sound. Please discuss this matter with scientists from the nursery/medical/healthcare domain.

The mHealth domain has been considered as a case study to show the applicability of the Wear-It API. The MoveUp app is a BCI module of a larger project, namely PERCIVAL. Parcival does not substitute the professionals involved in the medical treatment, but supports them in data collection. The suggestions are related to physical activity and the formulas have been validated thanks to the collaboration of experts (i.e. psychologists involved in Behavior Chang Interventions). I reduced the MoveUp description for sake of clarity. Anyway, I put a reference in the paper to give more information about the validation of the MoveUp approach according to experts’ knowledge.

Minor comments:

list of abbreviations line 524 is very useful. Please sort the entries in this table alphabetically.

Done. I removed some entries that were not necessary anymore.

In the abstract you mention that smartphones are equipped with many sensors, hereunder thermometers. Currently, I do not know of smartphones that are equipped with a thermometer sensor where the values can be used in apps. Thermometers might be available as external devices, but usually not built-in.

That’s right, sorry. I have revised the abstract to correct the mistake. Thank you.

paragraph starting line 60 is already a kind of conclusion in the introduction section. Please move to conclusion or similar.

The paper content has been completely revised. I hope it is clearer now.

line 69; style: "A massive work" --> "Much work"

Done.

line 70, style: avoid lengthy sentence in front of the references 5 and 6. why are these surveys well-known?

The surveys were cited by 4500 and 1180 papers respectively. Anyway, I removed the “well-known” definition.

Please avoid references as nouns; e.g., ref 23, etc. Further, remove "in" in front of references, e.g., line 107: "proposed [29]"; line 110: "Mobility in WSN is .... reasons [30] ..." and so on

Sorry, I didn’t understand this point. I left the references style unchanged.

line 185: Table with capital T

Done.

Table 1: the API seems rather small and straightforward. How does the API compare with other APIs?

Table 1 does not exist anymore. About API comparison, see the Discussion section and the points above.

Figure 3 is unclear.

Figures have been modified to make clearer their role in the paper.

line 289: why alphabetically ordered?

The paper content has been changed. I didn’t find this point.

line 302: Part with capital P

Done.

Figure 5: much space for little information ...

Figures have been modified to make clearer their role in the paper.

line 321: what do you mean by "profitably exploited" ? how is this measured?

The paper content has been changed. I didn’t find this point.

line 325: typo: word -> world

The paper content has been changed. I didn’t find this typo.

line 386: why is this UI sophisticated?

The discuassion about MoveUp VR has been modified.

The role of Section 4.1 and the scenario presented are unclear. How does this fit with the discussions around the API?

The paper content has been completely revised to make clearer the paper scope.

line 411: The role of virtual reality is unclear. Why are all these products/trademarks mentioned? This part reads more like a white-paper than a scientific article.

I reduced the part on MoveUp VR, putting it in the Discussion Section as an example of interoperability among apps at reasoning level of wearable environments.

line 434: references to SensorCap, Sensor Toolbox, LightBlue are not given.

Done.

line 452: Trough -> Through

Done.

line 466: unclear; what do you mean by "fully integrated"?

The MoveUp app can be installed within an application to allow it to use the API.

lines 478ff: references not given: PulseOn, NokiaSteelHR

Done.

line 489: strange sentence: why does an activity stem from a fact?

The paper content has been changed. I didn’t find this point.

Round 2

Reviewer 2 Report

The goals presented and the respected formulas are they validated? Are they part of a protocol of some sort?

Looking at EMBC conference I found this : Congestive Heart Failure Risk Assessment Monitoring through Internet of things and mobile Personal Health Systems. 40th Annual International Conference of the IEEE Engineering in Medicine and Biology Society (EMBC), 18-21 July 2018, Honolulu, HI, USA, 29 October 2018. IEEE (1557-170X/1558- 4615/https://doi.org/10.1109/EMBC.2018.8513024 ), doi: 10.1109/EMBC.2018.8513024. This is about using sensing devices and rule based IoT in order to manage heatlh – please ref, and extent related work of this topic.

The mHealth domain has been considered as a case study to show the applicability of the Wear-It API. I reduced the MoveUp description for sake of clarity. Anyway, I put a reference in the paper to give more information about the validation of the MoveUp approach according to experts’ knowledge (i.e. psychologists involved in Behavior Chang Interventions). 

Author Response

Dear reviewer,

I'm sorry, but I'm not able to find any new comments with respect to the first round. Your comments are extracted from the previous one, and my answer to your previous point has been reported too, as follows:

"The mHealth domain has been considered as a case study to show the applicability of the Wear-It API. I reduced the MoveUp description for sake of clarity. Anyway, I put a reference in the paper to give more information about the validation of the MoveUp approach according to experts’ knowledge (i.e. psychologists involved in Behavior Change Interventions)."

Thus, I'm not able to understand if you need further clarifications or not.

Reviewer 3 Report

The author has addressed many of my concerns. The main subject of this paper is now the API, as outlined in the title of the paper. However, the paper suffers from a lack of structure. As it is now, the paper presents some formal model (first in the conclusion named as "wearable environment notion") and a description of the API (which looks a bit like a programmer's manual). Further, there is no real research question (at least, I did not find one), nor a test/evaluation of hypotheses. There is a lengthy discussion, which is difficult to follow (i.e., difficult to extract an essence from; what is the main takeaway?).

What is the role of the wearable environment notion? Please name it consistently. How is this notion different from other notions?

Much of the new text is rather vague. e.g., line 292: "not necessarily". There are many sentences that are vague in that manner.

line 348: incomprehensible sentence. Please divide into several smaller sentences. Please also consider being more precise about what the article in Lancet refers to; specifically the last part "it continues ...".

line 363: incomprehensible: "About the MoveUp reasoning strategy ..."

line 385: vague: "a sort of middleware". What do you mean by "a sort of"? Please be more precise throughout the paper.

I find the discussion section is rather vague.

line 397: please remove "As reminded above"

line 404: unnecessary mention of examples, such as Empire State building and so on. Please remove "like the ..."

line 414 ff: incomprehensible. This is a scenario or example, but not something that fits into the discussion.

line 438: incomplete sentence / full stop at the wrong place.

line 442: reference to table missing.

line 447: how is the quality of data defined? How does your framework affect the quality of data? Do you have proof?

What is the main take-away from your paper? Please be more concrete about this.

other minor issues:

many typos; e.g.,

line 118 importabt->important

line 492 networf->network

Many sentences are quite long and difficult to understand; several of these sentences are probably not grammatically correct. Please review this from line 287 on and onward. e.g., sentence after semicolon line 288.

line 300: probably wrong font for "wd".

line 315, formula: (I suppose you use LaTeX) WEAR-IT appears as "WEAR minus IT_wd". Please use \mathrm{} for formulæ. the same at line 318. Also, other variables like WearOS and BLE might need different typesetting. Line 319: wd in the wrong font ... and so on.

line 334, and others: Step 4 with capital S

You refer quite often to "in the figure". Please use "in Figure x" instead. It is quite unclear which figure you mean.

line 373: Algorithm 1 with capital A

line 380: what do you mean by "the MoveUp one"?

Author Response

Dear Reviewer,

thank you very much for your valuable comments. I tried to do my best to take care of them in this revision of my paper. I hope you will appreciate this effort.

In the following, I explain how your issues have been tackled, point by point.

"The author has addressed many of my concerns. The main subject of this paper is now the API, as outlined in the title of the paper. However, the paper suffers from a lack of structure. As it is now, the paper presents some formal model (first in the conclusion named as "wearable environment notion") and a description of the API (which looks a bit like a programmer's manual). Further, there is no real research question (at least, I did not find one), nor a test/evaluation of hypotheses. There is a lengthy discussion, which is difficult to follow (i.e., difficult to extract an essence from; what is the main takeaway?).

What is the role of the wearable environment notion? Please name it consistently. How is this notion different from other notions?"

-> I have changed the abstract to explain how the paper focus is the definition of a framework for the development of expert systems capable to exploit wearable technologies. This framework, namely the wearable environment, has been thought to guarantee interoperability among applications and reliability of data.  To this aim, in the case study section, a concrete sample of the theoretical model presented in Section 3 has been proposed (see Figure 6) where the different components of the Wearable Environment have been practically and briefly characterized. Also the Discussion section has been completely revised to address the attention on interoperability and reliability. About the last point, the results of an experiment on data stored have been inserted to show how the wearable environment framework allows the development of algorithms to increase the quality of data managed by applications. Moreover, the comparison with similar tools has been reduced and moved to the conclusions, since it was not coherent with the discussion anymore. I hope this changes will answer your questions.

"

Much of the new text is rather vague. e.g., line 292: "not necessarily". There are many sentences that are vague in that manner.

line 348: incomprehensible sentence. Please divide into several smaller sentences. Please also consider being more precise about what the article in Lancet refers to; specifically the last part "it continues ...".

line 363: incomprehensible: "About the MoveUp reasoning strategy ..."

line 385: vague: "a sort of middleware". What do you mean by "a sort of"? Please be more precise throughout the paper."

-> I have tried to follow your suggestions, shortening longer sentences, correcting mistakes and so on. The API presentation has been left as it was, but many typos have been corrected. 

"

I find the discussion section is rather vague.

line 397: please remove "As reminded above" -> Done

line 404: unnecessary mention of examples, such as Empire State building and so on. Please remove "like the ..." -> Done

line 414 ff: incomprehensible. This is a scenario or example, but not something that fits into the discussion.

-> I left that part, since in my opinion it is useful to link the path definition algorithm (and the consequent use of MET value) and the introduction of the subsection, where it is specified that this function is useful for the MoveUp users, who are characterized by many difficulties when accomplishing PA.   

line 438: incomplete sentence / full stop at the wrong place. -> This part has been deleted

line 442: reference to table missing. -> This part has been deleted

line 447: how is the quality of data defined? How does your framework affect the quality of data? Do you have proof?

What is the main take-away from your paper? Please be more concrete about this."

-> The discussion section has been completely revised to focus the attention on interoperability and reliability issues. About the last point, the results of an experiment on data stored have been inserted to show how the wearable environment framework allows the development of algorithms to increase the quality of data managed by applications. Moreover, the comparison with similar tools has been reduced and moved to the conclusions.

"other minor issues:

many typos; e.g.,"

line 118 importabt->important -> Done

line 492 networf->network -> Done

Many sentences are quite long and difficult to understand; several of these sentences are probably not grammatically correct. Please review this from line 287 on and onward. e.g., sentence after semicolon line 288. 

-> I have reviewed the text to shorten longer periods and sentences. 

line 300: probably wrong font for "wd".

-> I left the font Italic to highlight that wd is a variable for the algorithm. 

line 315, formula: (I suppose you use LaTeX) WEAR-IT appears as "WEAR minus IT_wd". Please use \mathrm{} for formulæ. the same at line 318. Also, other variables like WearOS and BLE might need different typesetting. Line 319: wd in the wrong font ... and so on.

-> I corrected all the instances. Thank you!

line 334, and others: Step 4 with capital S

-> I have corrected all the instances of "step" in "Step"; thank you.

You refer quite often to "in the figure". Please use "in Figure x" instead. It is quite unclear which figure you mean.

-> Done.

line 373: Algorithm 1 with capital A

-> Done

line 380: what do you mean by "the MoveUp one"?

-> I have changed the sentence, thank you